# Research Progress on Sound Absorption of Electrospun Fibrous Composite Materials

**DOI:** 10.3390/nano12071123

**Published:** 2022-03-29

**Authors:** Xiuhong Li, Yujie Peng, Youqi He, Chupeng Zhang, Daode Zhang, Yong Liu

**Affiliations:** 1School of Mechanical Engineering, Hubei University of Technology, Wuhan 430068, China; 20200005@hbut.edu.cn (X.L.); pengyj1024@163.com (Y.P.); heyouqi1995@163.com (Y.H.); zcp1988123@126.com (C.Z.); 2Beijing Key Laboratory of Advanced Functional Polymer Composites, College of Materials Science and Engineering, Beijing University of Chemical Technology, Beijing 100029, China

**Keywords:** electrospun micro/nanofibers, sound absorption, porous materials, resonance

## Abstract

Noise is considered severe environmental pollutant that affects human health. Using sound absorption materials to reduce noise is a way to decrease the hazards of noise pollution. Micro/nanofibers have advantages in sound absorption due to their properties such as small diameter, large specific surface area, and high porosity. Electrospinning is a technology for producing micro/nanofibers, and this technology has attracted interest in the field of sound absorption. To broaden the applications of electrospun micro/nanofibers in acoustics, the present study of electrospun micro/nano fibrous materials for sound absorption is summarized. First, the factors affecting the micro/nanofibers’ sound absorption properties in the process of electrospinning are presented. Through changing the materials, process parameters, and duration of electrospinning, the properties, morphologies, and thicknesses of electrospun micro/nanofibers can be controlled. Hence, the sound absorption characteristics of electrospun micro/nanofibers will be affected. Second, the studies on porous sound absorbers, combined with electrospun micro/nanofibers, are introduced. Then, the studies of electrospun micro/nanofibers in resonant sound absorption are concluded. Finally, the shortcomings of electrospun micro/nano fibrous sound absorption materials are discussed, and the future research is forecasted.

## 1. Introduction

With the development of industrial production and urban construction, noise pollution is one of the four significant pollutants currently becoming an increasingly severe issue, highly endangering human health and restricting social development [1,2,3]. Therefore, controlling noise is essential, and using sound absorption materials is an efficient way to control noise pollution [4]. According to the sound absorption mechanisms, sound absorption materials can be classified as either porous or resonant sound absorption materials [5]. Porous sound absorption materials possess a large number of internal pores. When sound waves occur on their surfaces, friction between the materials and the air in the pores is generated. As a result, the sound energy will be converted into thermal energy and consumed due to the viscous and thermal effects. Resonant sound absorption materials are efficient for absorbing sound waves at low frequency, which are the equivalent of Helmholtz resonator, and consume sound energy by the resonance effect [6,7]. Porous sound absorption materials exhibit a good sound performance at high frequencies but perform poorly in the low and medium frequencies. However, resonant sound absorption materials show an opposite sound performance compared with the former. In addition, these materials also show disadvantages such as narrow sound absorption bands and bulky structures that need to be compounded with other materials to obtain a broader sound absorption band and a better sound absorption performance at high frequencies [8].

Low-frequency sound waves can easily bypass obstacles and are difficult to be absorbed by traditional sound absorption materials [9]. However, nanofibers with a high specific surface area can efficiently absorb the low-frequency sound waves. The large specific surface area of electrospun micro/nanofibers promotes the collision between the fibers and sound waves at low and medium frequencies, while the internal interconnected pores allow the diffusion of sound waves through the material, both of which boost the consumption of acoustic energy. Electrospinning is a technique for producing fibers with a diameter ranging from micron to nano scales, and it has been recently used to produce sound absorption materials [10,11,12].

Electrospun micro/nanofibers possess many advantages, such as small diameter, light weight, high specific surface area, and porosity. The sound absorption materials made of electrospun micro/nanofibers are promising for noise reduction, especially in traffic [13,14,15]. The sound absorption mechanism of electrospun micro/nanofibers is complex, and their sound absorption performance is mainly affected by several factors: material, morphology, thickness, spatial structure, and cavity depth. The type of materials is related to the properties of electrospun micro/nanofibers, such as piezoelectricity [16,17,18] and viscoelasticity [19], which influence the acoustic loss. Electrospun fiber properties such as fiber diameter [20], surface density [21,22], and microstructure [23,24] play essential roles in the process of sound absorption. The increase in the thickness of electrospun micro/nanofibers will not only lengthen the pore channels inside the material, but also enlarge the contact area between the material and acoustic waves, making it easier to consume acoustic energy [25]. Compared with 2D electrospun fiber membranes, three-dimensional (3D) electrospun fibers possess a larger surface area, manageable pore size, and mesoporous channels [26]. These features promote the consumption of sound energy, resulting in better sound absorption capacity of the 3D electrospun fibers [27,28]. Furthermore, setting a cavity behind the electrospun micro/nanofibers can form a Helmholtz resonator structure, improving their resonant sound absorption capacities [29,30].

Fibers fabricated by electrospinning have micro and nanoscale diameters, which provide high specific surface area and porosity. These features of electrospun micro/nanofibers lead to good sound absorption performance in low and medium frequencies [31]. In this review, we provide a focus on factors related to the sound absorption capacities of electrospun micro/nanofibers and their applications. Figure 1 presents an outline of this paper. First, the principles of how the materials and properties of electrospun micro/nanofibers affect the sound absorption performance are discussed. Second, recent research on electrospun micro/nanofiber sound absorption materials, which are classified as porous or resonant, is summarized. Finally, we outline the existing disadvantages of sound absorption materials produced by electrospinning and the prospects for their development.

## 2. Factors Influencing Sound Absorption Performance of Electrospun Micro/Nanofibers

Micro/nanofibers possess characteristics which are beneficial to the absorption of sound waves, namely small diameter, large specific surface area, and high porosity. However, micro/nanofibers usually have thin thicknesses, resulting in poor sound absorption. Micro/nanofibers are commonly combined with traditional sound absorption materials to achieve a better sound absorption effect. Electrospinning, one of the main methods of preparing micro/nanofibers, is divided into two types: solution and melt electrospinning [32]. There are significant limitations to melt electrospinning, such as a larger fiber diameter, the relative complexity of the design, and complicated devices [33]. Therefore, most researchers choose to fabricate micro/nanofibers by solution electrospinning.

The factors that influence the sound absorption performance of electrospun micro/nanofibers involve parameters such as materials, fiber diameter, surface density, microstructure, and thickness. Materials determine the properties of electrospun micro/nanofibers, such as piezoelectric and elastic properties. The fiber diameter, surface density, and microstructure of the fibers are related to the solution properties, process parameters, and environmental parameters of the electrospinning technology [34]. The thickness of electrospun micro/nanofiber membranes can be controlled by adjusting the electrospinning duration. Table 1 represents the recent research of the sound absorption coefficient (SAC) and properties of electrospun micro/nanofibers. The sound absorption capacity of the materials can be reflected by SAC, which is the ratio between the energy absorbed by the material and the total energy of the incident sound wave.

**Table 1 nanomaterials-12-01123-t001:** Summary of the properties and sound absorption performance of electrospun micro/nanofibers.

Materials	Fiber Diameter and Surface Density	Microstructure	Thickness of Nanofibrous Structures	SACs	Reference
Polyvinylidene fluoride (PVDF)/carbon nanotubes (CNTs) nanofibers and foam	138 ± 21 nm 45.5 g/m^2^	-	-	above 0.9 (1000 Hz)	[16]
PVDF/graphene (GP) nanofibers and acoustic nonwoven	169 ± 21 nm 45.5 g/m^2^	-	-	0.87 (1000 Hz) 0.95 (4000 Hz)	[17]
Nylon-6 nanofibers and Polyurethane (PU) foam	180 ± 10 nm	-	-	0.81 (600 Hz)	[35]
PU nanofibers and PU foam	300 ± 10 nm			0.59 (1700 Hz)	
PU nanofibers and polyethylene terephthalate (PET) nonwovens	509.9 nm	-	1.548 mm	0.9 (1800 Hz)	[36]
Polyacrylonitrile (PAN) nanofiber and spacer-knitted fabrics	110 ± 7 nm 17 g/m^2^	-	50 μm	0.7 (100–3200 Hz)	[37]
PAN/cellulose nanocrystals (CNC) aerogels	-	maze-like structure	50 mm	above 0.9 (600 Hz)	[38]
Poly (vinyl alcohol) (PVA) nanofibers	268 nm	Miura-ori structure	130 ± 5 μm	max value is 1.0	[39]
PU nanofibers and nonwovens	5–40 nm	nano-cobweb structure	1.2 mm	average value is 0.57	[40]
Polyvinylpyrrolidone (PVP) nonwoven mats of stacked nanofibrous layers	1.6/2.8 ± 0.5 μm 0.89 kg/m^2^	-	2.54 cm	max value is above 0.9	[41]
PAN nanofiber and perforated panel	333 ± 58 nm	-	205 ± 4 μm	max value is 0.93	[42]
Nylon-6 nanofibers with back cavity (30 mm)	150–200 nm	-	10 μm	0.6 (2000 Hz)	[43]

Acoustic models can be applied to theoretically evaluate the sound absorption properties of the materials, which are useful for directing the design of sound absorption materials. Recently, several acoustic models used for investigating the SAC of sound absorption materials have been published [39,44]. Previously, there were two types of acoustic models commonly employed [4]. The first is the empirical model with few parameters, which is relatively easy to establish. The most representative empirical model is the Delany–Bazley model, related to the airflow resistivity of the materials, but it can only be employed for predicting the acoustic behaviors of porous materials in the frequency range of 250–4000 Hz [45,46]. In order to obtain more accurate predictions, some studies made specific corrections to the Delany–Bazley model and developed several new models. For example, Miki proposed a modified expression, known as the Delany–Bazley–Miki model, based on the Delany–Bazley model [47,48]. The second acoustic model is the phenomenological model. Compared to the empirical models, it involves non-acoustical physical parameters and can improve the prediction accuracy. In terms of phenomenological model, the Johnson–Champoux–Allard (JCA) model is the most-used method for describing the sound propagation in sound absorption porous materials [49,50].

### 2.1. Materials

The sorts of materials determine the properties of electrospun micro/nanofibers. Modifying the materials by altering the material or adding other matter can change the physical and chemical properties of the fibers, thus affecting the sound absorption performance of the materials.

Polyvinylidene fluoride (PVDF) is a material with the piezoelectric property that can convert acoustic energy into electric energy, thus facilitating the absorption of acoustic energy [51]. Wu et al. [16,17,18] conducted a series of studies on electrospun PVDF nanofibers. They added carbon nanotubes (CNTs), GP, and silver nanoparticles (AgNPs) into PVDF solutions and fabricated them into nanofiber membranes by electrospinning. The results show that the addition of CNTs and GP increases the surface area and the β-phase crystallinity of PVDF nanofiber membranes, enhancing the contact between materials and sound waves and improving piezoelectricity, thereby promoting the absorption of sound waves at low frequency. PVDF/AgNPs nanofiber membranes have excellent piezoelectricity and acoustoelectric conversion characteristics. The composite nanofiber membranes can convert acoustic energy into other forms of energy and show great potential for sound absorption applications. Another group reported that electrospun polyacrylonitrile (PAN) nanofiber membranes display greater piezoelectric conversion capability compared to PVDF nanofibers [52]. For this purpose, Shao et al. [53] prepared an acoustoelectric device by interposing an electrospun PAN fibrous membrane to two metal-coated polyethylene terephthalate (PET) films. The electrospun PAN nanofiber membranes can convert noise into electric power at low and medium frequencies. Furthermore, they prepared a single-layer nanofiber membrane made of a PAN/PVDF polymer blend, which possesses an acoustoelectric energy conversion efficiency as high as 25.6%. Figure 2 shows the noise harvester structure [54].

The elasticity of the materials may influence the resonant process between electrospun micro/nanofibers and sound waves. Good elasticity of materials may promote the consumption of acoustic energy by the resonance effect, thus improving the sound absorption performance of the materials. Park et al. [35] prepared polyurethane (PU) and nylon 6 composite nanofibers by electrospinning and then laminated them with polyurethane foam. They found that PU composite nanofibers perform better in sound absorption than nylon 6 composite nanofibers. Furthermore, the PU nanofibers exhibit superior air permeability and elasticity, and the sound waves can easily propagate into the material, leading to the increased consumption of acoustic energy by vibration. On the contrary, some researchers believe that electrospun micro/nanofibers with less elasticity have more friction with sound waves, promoting efficient sound absorption. Rabbi et al. [55] demonstrated the effects of applying PU and PAN nanofibers within polyester and wool nonwovens. Owing to the higher air permeability and elasticity of PU compared with PAN nanofiber layers, the composite with PAN nanofibers possesses a higher sound transmission loss.

### 2.2. Diameter and Surface Density

Diameter and surface density are the critical factors affecting the flow resistivity and natural resonant frequency of the materials. Diameter impacts the specific surface area and porosity of electrospun micro/nanofibers, while the surface density determines the fibers’ distribution. The sound absorption capability of materials can be represented by flow resistivity, which reflects the extent of curvature on the path of sound waves traveling through the materials. Materials with a low natural resonant frequency usually exhibit good sound absorption capacity at low frequency. Thus, suitable diameter and surface density for electrospun micro/nanofibers enable good sound absorption. For instance, Akasaka et al. [44] prepared nonwoven sheets composed of thin silica fibers with diameters ranging from 0.72 to 3.44 μm by electrospinning. They found that the sound absorption performance of the silica fibers is related to a critical value (approximately 3 μm) of the fiber diameter. When the fiber diameter is above the critical value, the sound absorption coefficient of the composite increases with the decrease in the fiber diameter in a wide frequency range. The diameter and surface density of electrospun micro/nanofibers can be controlled by varying the parameters of the electrospinning process. Shou [20] evaluated the effects of the processing parameters such as solution concentration, voltage, collecting distance, the diameter of the sprayer, and environmental temperature on the diameter of the electrospun fibers. The diameter of electrospun fibers can be changed by adjusting the processing parameters mentioned above, thus altering the flow resistivity and sound absorption efficiency of the materials. Du [56] adopted a nano spider spinning device to fabricate PAN nanofiber membranes and controlled the diameter and surface density of the fiber membranes by adjusting the voltage and transmission speed of the receiving device. The sound absorption performance of the composites composed of electrospun nanofiber membranes and nonwoven fabrics was tested. The results demonstrate that the magnitude of fiber diameter and surface density is inversely proportional to voltage, within a certain range. The increment of the transmission speed of the collector decreases the surface density of the composite and increases its porosity. When the sound waves’ frequency is higher than 500 Hz, increasing the surface density of the nanofiber membrane or decreasing its diameter can improve the sound absorption coefficient of the composites. Kalinová [57] studied the effects of feeding rate and collecting distance on surface density and fiber diameter of electrospun nanofiber membranes. It was found that electrospun nanofiber membranes’ surface density decreases with the increase in feed rate, and the larger the receiving distance, the smaller diameter. In addition, the resonant frequency of the nanofiber membrane is inversely proportional to both the average fiber diameter and the surface density.

### 2.3. Microstructure

The microstructures of the electrospun micro/nanofibers influence the sound absorption performance of the materials. For example, the bead structure will increase the porosity, surface area, and bulkiness of the fiber, causing irregular changes in the fiber diameter. Moreover, the core-shell or hollow structure may influence the density and natural resonant frequency of the electrospun micro/nanofibers, therefore modifying their response to sound waves at different frequencies. Therefore, electrospun micro/nanofibers with different microstructures can be fabricated by adjusting the process parameters and devices of electrospinning, and their sound absorption performance will be influenced. Zhang [23] discussed the relationship between poly(lactic acid) (PLA) solution concentration and the morphology of the electrospun micro/nanofibers and observed that the number of bead structures decreases with the increase in solution concentration. When the PLA solution concentration is 13% and 14%, the electrospun nanofibers with the fine diameter and a few bead structures perform best in sound absorption. Similarly, Gao [58] fabricated polyvinyl alcohol (PVA) electrospun fiber mats with different bead morphologies by varying the solution concentration and investigated the sound absorption performance of the composites with nonwoven fabrics. The results indicate that the bead structures appear in the electrospun fiber mats with 3 wt.% to 7 wt.% concentration of PVA solution. The diameter and the number of the beads will increase as the concentration of PVA solution decreases. Yoon et al. [59] prepared a micro-glass bead/PLA porous fiber composite by electrospinning. However, the sound absorption of the porous fibers is lower than normal fibers because the porous structures do not change the void ratio of the composite, but have increased its natural frequency. The electrospun micro/nanofibers with unique microstructures can also be produced using specially designed electrospinning devices. For instance, the core-shell or hollow structure may influence the fibers’ density and resonance frequency, therefore modifying their response to different frequencies of sound waves. Bertocchi et al. [24] fabricated a core-shell fiber with polycaprolactone (PCL) as the surface layer and polyethylene glycol (PEG) as the core layer by using a coaxial electrospinning device that they installed themselves. Compared with single-phase fibers, the core-shell fiber structure exhibits a higher absorption capacity for all noises. The sound absorption ability of the core-shell fibers is proportional to the fluid viscosity of the core layer, and its sound absorption frequency range can be tuned by controlling the core layer’s fluid viscosity.

### 2.4. Thickness

The thickness of electrospun micro/nanofibers is a critical factor for their acoustic performance, affecting the propagation distance of sound waves within the fibers. Thickness variation of electrospun micro/nanofibers can be achieved by adjusting the electrospinning duration. Zou [60] reported that the sound absorption performance of PU and PVDF nanofiber mats prepared by electrospinning for 2 h is similar to traditional foam. As the electrospinning time increased to 4 h, the sound absorption performance of PU and PVDF nanofiber mats was significantly enhanced. The sound absorption coefficient of the samples is above 0.5 when the frequency is nearly 1000 Hz, reflecting the excellent sound absorption performance of electrospun fibers at low and medium frequencies. Salehi et al. [61] reported a study applying multi-layered PET nonwoven structures integrated with PAN nanofibers. They obtained different deposition amounts of composite fibers using three electrospinning times of 15, 60, and 180 min. They suggested that the sample with the electrospinning time of 60 min shows the best sound absorption performance. The increase in nanofiber deposition amount can improve the sound absorption coefficient of nanofibers, but the best sound absorption performance could not be achieved by increasing deposition time continuously. Similarly, Özkal et al. [36] produced new sound absorption materials by incorporating PU nanofibers of several electrospinning durations (5, 20, 60, and 120 min) with recycled PET bottle waste nonwovens. It is found that the resonant frequency of the composite decreases with the increase in the spinning duration. Ding et al. [40] prepared multi-layer PU nano-membranes with a nano-cobweb structure using a needleless electrospinning method. The nano-cobweb multi-layer material with an average sound absorption coefficient of 0.63, prepared by electrospinning for 240 min, dramatically improves the sound absorption performance over the multi-layer. Avossa et al. [41] obtained polyvinylpyrrolidone (PVP) nonwoven mats of stacked nanofibrous layers by electrospinning that yielded reduced thickness and excellent sound absorption properties in the low and medium frequency range. The PVP mats’ sound absorption coefficient is much higher than 0.9 at 450 Hz, and its sound absorption performance can be continuously tuned by changing the mass. Ji et al. [62] designed a sound absorber made of electrospun poly (vinylidene fluoride-co-hexafluoropropylene) (PVDF-HFP) fibrous membrane and melamine foam. The result shows that the sound absorption performance of the sound absorber is significantly affected by the thickness of the two materials.

## 3. Electrospun Micro/Nanofiber-Based Porous Sound Absorption Materials

Porous sound absorption materials have many internal pores, which are beneficial for sound waves to enter the interior of the materials and consume energy by friction. The sound absorption principle of porous sound absorption materials is shown in Figure 3 [63]. Porous sound absorption materials perform well in sound absorption and possess advantages of low cost, easy formation, and light weight. Porous acoustic materials are ideal noise reduction materials that can reduce noise in construction and transportation fields [4]. Foams and fibers are commonly used as porous sound absorption materials. Both possess high sound absorption coefficients at the high-frequency range, but show poor sound absorption at the low and medium frequencies [44]. The poor sound absorption performance of porous sound absorption materials in the low and medium frequencies limits their applications. To broaden the applications of porous sound absorption materials, researchers combine them with electrospun micro/nanofibers, which improves the sound absorption performance by expanding the contact area of the material with acoustic waves and enhances the material properties such as water resistance, high-temperature resistance, and mechanical strength. Furthermore, the addition of lightweight nanofibers will not vastly increase the weight and size of the sound absorption materials [43]. For instance, combining electrospun micro/nanofibers with traditional sound absorption foams and fibers can increase the contact area between the materials and sound waves, thus improving the sound absorption performance at low and medium frequencies. Moreover, electrospinning micro/nanofibers with special structures can be prepared by the refinement of electrospinning devices and material modifications, which shows promising application prospects with a good sound absorption performance at low and medium frequencies.

### 3.1. Composite of Sound Absorption Foam and Electrospun Micro/Nanofibers

Foams are typically porous materials with good sound absorption properties, and they have been widely used for sound absorption in the field of transportation [64]. The cellular morphology of foams includes a cavity with three pores (open, partially open, and closed) [65]. Open-cell foams have many interconnected pores, while the pores of the closed-cell foams are isolated from each other. The type of pores affects the sound absorption properties of the foams; the open-cell foams are more efficient than closed-cell foams in sound absorption [66]. The sound absorption principle of partially open-cell foams is similar to that of open-cell foams, but their sound absorption capability is inferior. Therefore, open-cell foams are the primary sound absorption foam materials.

Traditional foam sound absorption materials have large pore sizes and low porosity. The sound waves cannot vibrate sufficiently inside the pores, leading to the poor sound absorption performance of the foams at low frequency. Increasing the thickness of the material or setting a cavity behind the foam are common methods used to promote their sound absorption performance at low frequency. However, these methods will cause the waste of material and space. Laminating electrospun micro/nanofibers with foam materials can enhance the sound absorption performance of foams at all frequencies, especially at low frequencies [67]. Zou et al. [60] made three composite nanofiber mats by solution electrospinning of PU, PVDF, and PU/PVDF blends on the surface of porous foams and examined their sound absorption performance. The sound absorption coefficients of the composites are higher than the typical acoustic foam in the frequency range of 800–1600 Hz. The composite made of foam and PU/PVDF nanofibers has the highest absorption coefficient of nearly 0.7. Tomáš [68] found that the 10 mm-thick foam treated with electrospun nanofibers and the untreated 20 mm-thick foam sheets exhibit similarly shaped absorption coefficient curves in a wide frequency range. The foam combined with nanofiber membrane shows a higher absorption coefficient in the 1500–4000 Hz frequency range. The above studies suggest that composite performances of electrospun micro/nanofibers and foam acoustic materials are better than those of the typical foam acoustic materials.

The sponge is a type of acoustic foam with many applications in noise reduction. Still, its substantial water absorption property in wet environments leads to a poor sound absorption performance. Using electrospun micro/nanofibers to modify the sponge can improve its sound absorption performance and reduce the influence of the environment on sound absorption [35]. Xiang et al. [69] covered the sponge surface with electrospun polymethylsilsesquioxane (PMSQ). The addition of PMSQ enhances the sponge’s sound absorption performance, and the resulting composite shows a strong water resistance property. As a result, the composite can be used for noise reduction in aqueous environments. Cao [70] fabricated ultralight polystyrene (PS) nanofiber sponges with excellent sound absorption properties and hydrophobicity using the humidity-assisted electrospinning method. The PS sponges significantly improve the sound absorption coefficients when compared to those of nonwovens and melamine foams in the frequency range of 500–1000 Hz. Furthermore, the PS sponges exhibit durable sound absorption performance at different humidity levels.

### 3.2. Composite of Sound Absorption Fibers and Electrospun Micro/Nanofibers

Fibrous structures can absorb, reflect, and transmit the incident sound waves simultaneously [71]. According to their compositions, acoustic fiber materials can be classified into natural, synthetic, and metal fibers. Natural and synthetic fibers are widely used due to their low price and superior sound absorption performance at high frequency. Their sound absorption performance at the low and medium frequencies can be effectively improved by compounding with electrospun fibers [72,73].

Composites of electrospun micro/nanofibers and natural fibers have improved sound absorption performance while retaining the environment-friendly and biodegradable features [74]. For instance, Selvaraj et al. [75] formed a composite by covering modified PVA electrospun nanofibers with coir fibers. Figure 4 shows the preparation process and the sound absorption performance of the samples (coir, PVA nanofibers/coir, and Fleshing hydrolysate (FH)/PVA nanofibers/coir). The composite shows a higher sound absorption coefficient in the frequency range of 400–1000 Hz and a lower frequency of sound absorption peaks compared with coir fibers. Na et al. [76] laminated melt electrospun nylon microfibers with wool fabric. The composite performs better than knitted wool fabric in sound absorption at frequencies between 1000 and 4000 Hz, with the highest sound absorption coefficient of 0.85 at 4000 Hz. Ozturk et al. [77] coated electrospun PAN nanofiber membranes on the surface of wool and jute fiber felts, respectively, and inspected their acoustic properties. The results suggest that the addition of nanofiber membranes would change the permeability of the fiber mats, further affecting the sound absorption performance. The sound absorption performance of wool felt is improved by four times after being composited with the PAN nanofibers. The sound absorption coefficient of jute felts could be improved at least 4 to 8 times, and its maximum could reach 0.4 in the frequency between 500 and 1500 Hz by coating nanofibers. In another work, an alternative sound absorption material was designed [78]. The material is composed of nanofiber resonant membranes together with wool felts covering polypropylene (PP) nonwovens, which displays better sound absorption properties than wool felts alone and is suitable for sound absorption on white goods.

Synthetic fibers are extensively applied in sound absorption fields due to their high performance and economical price [1,79]. The sound absorption performance of synthetic fibers at low frequency can be improved by combining them with electrospun micro/nanofibers [9,30]. For instance, electrospun nanofibers laminated with nonwoven fibers can effectively improve the sound absorption coefficient of the materials, and the composites display higher absorption coefficients when the nanofiber side is facing the sound source [23,56]. Yang et al. [25] prepared reduced graphene oxide (RGO)/PAN nanofiber nonwovens by applying electrospinning and bi-component melt-blown nonwovens technology. In the range of 500 to 6300 Hz, the sound absorption performance of the composites increased with the addition of RGO, and the sound absorption coefficient can increase by 24%. Liu et al. [80] prepared PVA/polyethylene oxide (PEO) electrospun nanofiber membranes with graphene oxide (GO) of various concentrations and laminated them with nonwoven fabrics. The addition of nanofibers significantly improves the sound absorption performance of the materials at a low frequency. Ozturk et al. [37] fabricated a nanofibrous coated composite consisting of electrospun PAN nanofibers and polyester spacer warp-knitted fabrics. By adjusting the deposition amount and surface coating arrangement, the sound absorption coefficient of the composite can reach 0.7 at low and medium frequencies. Özkal et al. [81] reported a sound absorption material made of electrospun PU nanofibers and polyester needle-punched nonwovens. This composite shows an excellent improvement in sound absorption compared with commercial sound absorption materials. Karaca et al. [82] covered thermoplastic polyurethane (TPU) and TPU/PS sub-micro fiber webs on the surface of rigid glass fiber fabric reinforced epoxy composites (GFEC) and flexible PP spun bond nonwovens. The sound absorption coefficient of GFEC is improved from 0.1 to 0.4 by combining them with sub-micro web layers with a thickness of less than 1 mm, and the maximum sound absorption coefficient of the PP nonwovens moves to a lower frequency after being covered by sub-micro fiber webs.

### 3.3. Electrospun Micro/Nanofiber Based Sound Absorption Materials

The high porosity and specific surface area of electrospun micro/nanofibers increase the contact between the material and sound waves. However, electrospun micro/nanofibers are usually deposited as thin mats or membranes, making them perform poorly in sound absorption. Thus, they cannot be used for noise reduction individually. For this purpose, researchers have investigated the development of new sound absorption materials made of electrospun micro/nanofibers.

Electrospun micro/nanofibers can be made into excellent sound absorption materials by changing their microscopic or spatial structure. Chang et al. [27] electrospun a nanofiber with a 3D lattice structure by using two syringes with opposite polarities, oppositely placed. The sound absorption performance of electrospun three-dimensional nanofiber is higher than that of the commercial acoustic cotton in the range of 400 to 900 Hz, and its sound absorption coefficient is up to 0.9. Selvaraj et al. [83] prepared a nanofiber layer using electrospun PVA and collagen hydrolysate extracted from waste leather trimmings. The layer is sandwiched between PAN nanofibrous layers. The composite structure is porous and shows good sound absorption performance in the 800–2500 Hz frequency range. Zong et al. [84] reported a fibrous sound absorption sponge with an interlocked dual-network-induced stable fluffy-stacked structure formed by PSU microfiber and PVDF nanofiber networks. The structure gives the fibrous sponge an excellent sound absorption property at low frequency, and its sound absorption coefficient could reach 0.93 at 1000 Hz. Similarly, Feng et al. [85] produced gradient structured fibrous sponges (GSFSs) with high sound absorption capacity in a wide frequency range by combining humidity-assisted multi-step electrospinning and physical/chemical double cross-linking techniques. Its fabrication process is shown in Figure 5a, and its sound absorption is illustrated in Figure 5b.

The sound absorption performance of electrospun micro/nanofibers can also be influenced by adding fillers in the polymer solutions. Elkasaby et al. [86] reported that the sound absorption coefficient of electrospun polyvinyl chloride (PVC) nanofibers is influenced by adding CNTs. The addition of 5 wt% CNTs in PVC nanofibers increases the sound absorption by 62% compared with PVC electrospun nanofibers, and the sound absorption coefficient of the composite reaches the maximum value of 0.4 at 800 Hz. Sorbo et al. [87] found that the addition of graphene can affect the sound absorption properties of electrospun PVP mats in a non-monotonous manner.

Recently, sound absorption materials made of aerogels have been reported. Aerogels are mesoporous and open-cell solids with high porosity and low density [88,89]. The open-cell structure of aerogels is beneficial to increase the friction and collision between sound waves and materials in the transfer process of sound energy, endowing the aerogels with good sound absorption performance [28]. Ding and his colleagues have produced a series of nanofibrous aerogels by electrospinning. Si et al. [90] produced fibrous, isotropically bonded elastic reconstructed (FIBER) nanofibrous aerogels (NFAs) with 3D structure that are highly compressible and resilient using electrospinning and fibrous freeze-shaping techniques. The sound absorption coefficient of this material is higher than that of commercial non-woven materials at the frequency range from 100 to 6300 Hz (shown in Figure 6a–c). Cao et al. [91] designed nanofibrous aerogels with bamboo lashing-like structures through a freeze-drying method (shown in Figure 6d–f). Benefiting the internal hierarchical porous structures and hydrophobicity, the nanofibrous aerogels exhibit excellent moisture resistance, are ultralight, and show efficient sound absorption capability with a noise reduction coefficient (NFC) of 0.41. Cao et al. [38] fabricated composite nanofiber aerogels with a hierarchical maze-like microstructure by weaving cellulose nanocrystal lamellas with electrospun PAN nanofibers through the freeze-casting technique (shown in Figure 6g–i). The composite aerogels exhibit a remarkable sound absorption ability due to the maze-like structure that increases the contact area between the sound waves and the composite. Compared to the fiber aerogels with a network structure, the designed maze-like structure significantly enhances sound absorption capacity in the low-frequency range, and its NFC can reach 0.58.

## 4. Electrospun Micro/Nanofiber Based Resonant Sound Absorption Materials

Resonant acoustic materials have a high absorption coefficient at low frequencies, making up for the lack of sound absorption capacity of porous acoustic materials at low and medium frequencies [92]. However, the sound absorption frequency range of traditional resonant acoustic materials, such as perforated panels and microperforated panels, where the thin plate is narrow, exhibits poor individual sound absorption effects. To achieve a better sound absorption effect, they are usually combined with other acoustic materials. Figure 7 shows a perforated porous material [93]. Electrospun micro/nanofibers yield both porous and resonant sound absorption capability. They can improve the sound absorption performance of traditional resonant sound absorption materials and can be used as a resonant acoustic material to absorb sound waves at low frequencies.

Traditional resonant acoustic materials show narrow sound absorption bands. However, decorating them with electrospun micro/nanofibers can effectively broaden the absorption frequency range and improve their sound absorption coefficients. Guo et al. [94] prepared PVP nanofiber membranes using the electrospinning technique with a microperforated panel as a collector to deposit the fibers. The microperforated panel and obtained nanofibers form a composite sound absorption structure. The composite establishes a wider sound absorption frequency range than the single-layer microperforated panel, and its absorption coefficient is higher at the frequency range of 1500 to 2500 Hz. However, the sound absorption coefficient of the composite decreases from 2500 to 6000 Hz, and its sound absorption frequency range becomes narrow when the thickness of the nanofiber surpasses a certain value. Therefore, the thickness of the nanofiber membranes needs to be controlled within a certain range to attain an excellent sound absorption performance at low frequency. Tomáš et al. [68] adopted electrospun PVA nanofiber membranes to modify the perforated panel and investigated the effect of the back cavity depth on the sound absorption performance of the composite. The results reveal that the sound absorption peak of the composite shifts to a low frequency with the increase in the back cavity depth. When the depth of the back cavity is 34 mm, and the surface density of the nanofiber membrane is 0.6 gsm, the sound absorption coefficient of the composite material can reach 0.7 in the 300 to 400 Hz frequency range. Xiang et al. [42] fabricated three electrospun PAN nanofiber membranes with different thicknesses and porosities. They studied the sound absorption properties of the composites made of different nanofiber membranes and perforated panels, respectively. The results illustrate that the addition of a nanofiber membrane substantially improves the sound absorption performance of the perforated panel. When the depth of the back cavity is 10 mm, the highest sound absorption coefficient of the composite can reach 0.93, which is 3 times the absorption coefficient of a single perforated panel.

Although composites made of electrospun micro/nanofibers and traditional resonant acoustic materials have a good sound absorption performance, they still have problems such as the large size and bulky structure, causing the waste of space and limiting their applications. Using electrospun micro/nanofibers as resonant acoustic materials individually is an effective solution to save space, expanding the applications of resonant acoustic materials. The Helmholtz resonator is a structure that absorbs sound waves by resonance, and electrospun micro/nanofiber membranes can form this structure by incorporating a cavity. Therefore, the sound absorption performance of the electrospun fiber membranes at the low and medium frequencies can be promoted by setting an air layer between the fiber membrane and the rigid wall [30]. Liu et al. [29] compared the effects of different cavity depths on the sound absorption performance of three electrospun nanofibers of PAN, TPU, and thermoplastic polyester elastomer (TPEE). It is shown that the sound absorption coefficients of the nanofiber membranes increase in the frequency range from 100 to 1000 Hz as the cavity depths increase. Wang et al. [39] fabricated a PVA nanofibrous membrane with a Miura-ori sandwich structure by integrating electrospinning and paper folding techniques, and its fabrication process is shown in Figure 8. The sound absorption coefficient of Miura-ori sandwich metamaterial is improved to 1.0, and the first resonance frequency shifts to the lower frequency range compared with flat nanofiber membranes.

Furthermore, nanofiber membranes can vibrate at a low frequency when forced by sound waves, and the energy of sound waves will be absorbed by the principle of membrane resonance [95,96,97]. Kalinová et al. [98] designed a resonant structure with a nanofiber membrane, grid frame, and cavity based on the resonance principle. The nanofiber membrane converts the acoustic energy into kinetic energy, and the metal frame promotes energy dissipation. The size of the grid frame determines the resonant frequency of the structure, and the absorption coefficient is related to the depth of the cavity. When the grid size is 4.6 × 3.8 mm, and the cavity depth is 50 mm, the structure exhibits the best absorption performance, with the highest absorption coefficient of 1 at the frequency range between 600 and 700 Hz.

The optimization of the electrospun micro/nanofibers’ structure is a way to enhance their resonant sound absorption ability, which can turn them into a sort of resonant sound absorption material. Xu et al. [99] produced nanoporous microspheres by electrospinning. These microspheres can be deemed an assembly of several thousand Helmholtz resonators, absorbing low-frequency sound waves by resonance. Shen et al. [100] reported a flexible PVA nanofiber microperforated structure prepared using electrospinning and a punching process (shown in Figure 9). The microperforated membrane possesses both porous and resonant sound absorption effects, and it can be used individually.

The sound absorption frequency range of traditional resonant acoustic materials can be broadened by compounding them with electrospun micro/nanofibers. Yet, their sizes are still too large for commercial applications. Resonant acoustic materials manufactured with electrospun micro/nanofibers possess less volume, and their acoustic properties can be controlled by adjusting electrospinning parameters. Therefore, it is necessary to study the resonant sound absorption materials prepared by electrospinning.

## 5. Conclusions

This review introduces the recent research of electrospun micro/nanofibers for sound absorption. Electrospun micro/nanofibers have advantages in sound absorption owing to their small fiber diameter, high specific surface area, and high porosity. Their sound absorption properties are determined by factors such as material, diameter, surface density, microstructure, and thickness, which can be controlled by changing the materials, parameters, and duration of electrospinning. Electrospun micro/nanofibers have various forms of sound absorption. The composite of electrospun micro/nanofibers and porous materials are effective in sound absorption at medium and high frequencies. Decorating resonant sound absorption materials with electrospun micro/nanofibers can enhance their sound absorption properties at a low frequency. In addition, electrospun micro/nanofibers with a 3D or Helmholtz resonator structure can also be deemed as sound absorption materials by themselves.

In the future, electrospun micro/nanofibers will surely be promising materials for the application of sound absorption in many fields. However, the application of electrospun micro/nanofibers for sound absorption faces several challenges. First, the sound absorption properties of electrospun micro/nanofibers and their composites at low frequency require additional improvement. Theoretical research can reduce the cost and pollution caused by the experimental process, and it can study the factors affecting the sound absorption performance of electrospun micro/nanofibers. The sound absorption performance of materials can be predicted by establishing a sound absorption model. The accuracy of the prediction can be increased by modifying the model according to the material and structure of the electrospun micro/nanofibers. Second, traditional electrospun micro/nanofibers have many limitations in its application under several different conditions, such as moist and high temperature environments, owing to their poor thermal stability, water resistance, and mechanical properties. Therefore, the multifunctional modification of the fibers is necessary. Electrospun micro/nanofibers can obtain properties such as good water resistance and thermal stability by modifying the materials or preparing them using an innovative process, which is beneficial for broadening their application for sound absorption. Finally, most electrospun micro/nanofibers are fabricated by solution electrospinning, which exhibits the problem of low efficiency, and the use of solvents is usually corrosive and toxic. These characteristics of solution electrospinning are not beneficial to the industrial production of electrospun micro/nanofibers and may lead to environmental concerns. Hence, it is urgent to find ways to solve the problems inherent to solution electrospinning. Using water-soluble polymers in solution electrospinning or choosing eco-friendly melt electrospinning to replace the former in the future are suggestions for solving these problems. In conclusion, further exploration of electrospun micro/nanofibers will promote their sound absorption applications in many fields.

## Figures and Tables

**Figure 1 nanomaterials-12-01123-f001:**
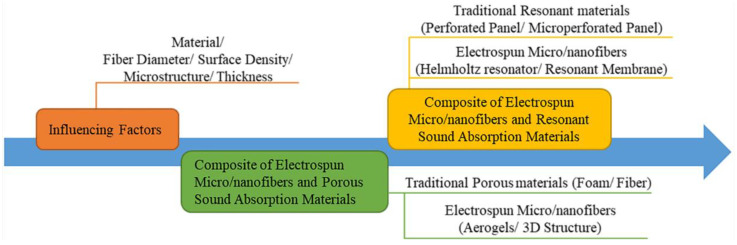
The outline map of this review.

**Figure 2 nanomaterials-12-01123-f002:**
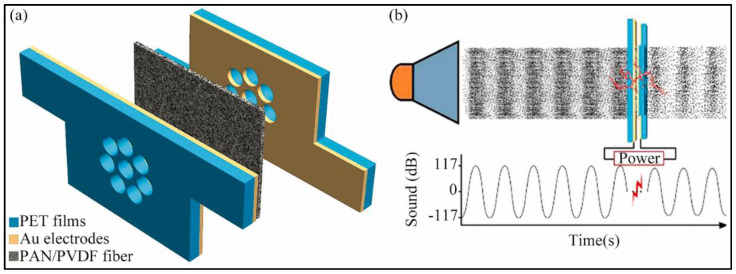
(**a**) Schematic illustration of the PAN/PVDF noise harvester structure; (**b**) Principle of sound energy harvest. Reprinted with permission from ref. [54]. Copyright 2021 Elsevier, Amsterdam, The Netherlands.

**Figure 3 nanomaterials-12-01123-f003:**
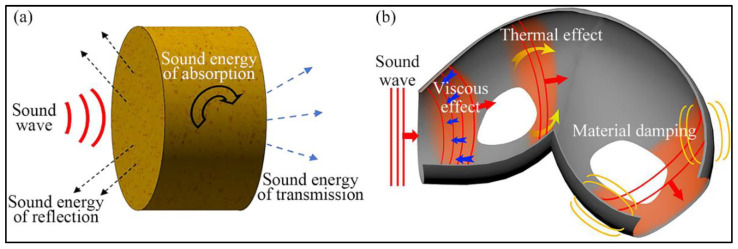
(**a**) Schematic illustration of the sound absorption process for porous materials; (**b**) Process of the sound energy consumption in the porous sound absorption materials. Reprinted with permission from ref. [63]. Copyright 2017 American Institute of Physics, College Park, MD, USA.

**Figure 4 nanomaterials-12-01123-f004:**
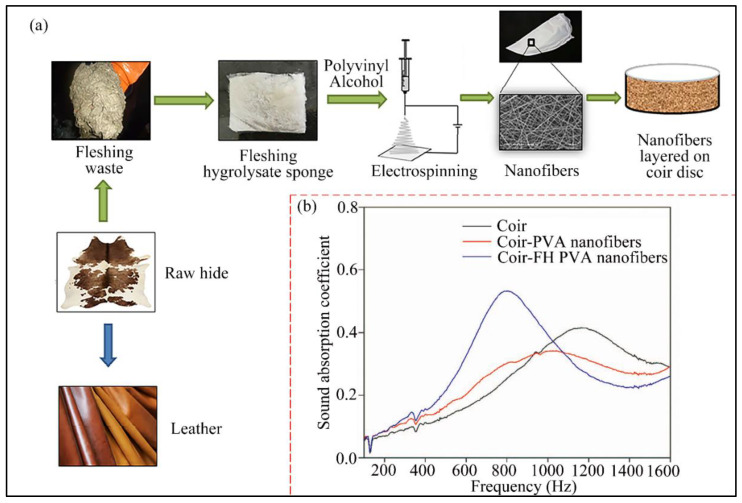
(**a**) The preparation process of nanofibers and coir composites by electrospinning; (**b**) The sound absorption coefficients of the samples. Reprinted with permission from ref. [75]. Copyright 2019 Elsevier.

**Figure 5 nanomaterials-12-01123-f005:**
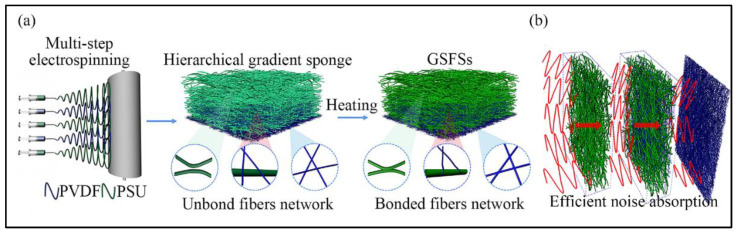
Fabrication of GSFSs (**a**) Fabrication process of GSFSs; (**b**) Sound absorption process of GSFSs. Reprinted with permission from ref. [85]. Copyright 2021 Elsevier.

**Figure 6 nanomaterials-12-01123-f006:**
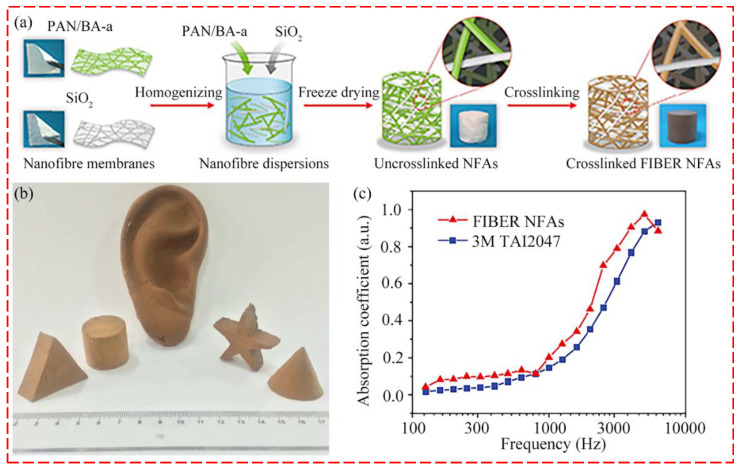
Several nanofiber aerogels for sound absorption. (**a**) Synthetic steps of FIBER NFAs; (**b**) Photographs of FIBER NFAs with several shapes; (**c**) Sound absorption coefficients of FIBER NFAs and commercial 3M TAI2047 nonwovens. Reprinted with permission from ref. [90]. Copyright 2014 Springer Nature. (**d**) Fabrication process of RNFAs; (**e**) Microstructure of RNFAs; (**f**) Sound absorption coefficient of RNFAs with different densities. Reprinted with permission from ref. [91]. Copyright 2019 Royal Society of Chemistry. (**g**) Fabrication of NFAs; (**h**) Sound absorption mechanism of PAN-CNC40; (**i**) Sound absorption coefficients of the PAN-CNC40 and the commercial fibrous sound absorption felt. Reprinted with permission from ref. [38]. Copyright 2021 Elsevier.

**Figure 7 nanomaterials-12-01123-f007:**
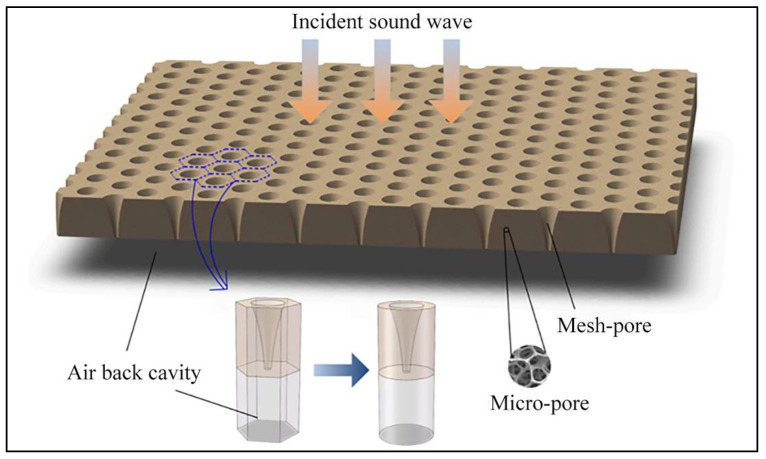
Schematic illustration of a gradually perforated porous material backed with a Helmholtz resonator. Reprinted with permission from ref. [93]. Copyright 2021 Elsevier.

**Figure 8 nanomaterials-12-01123-f008:**
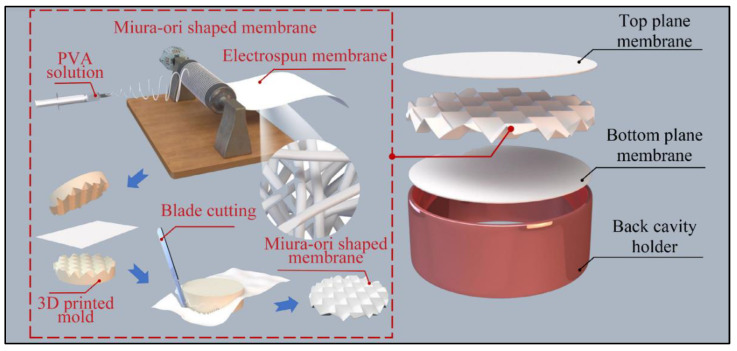
Fabrication of the Miura-ori sandwich structure. Reprinted with permission from ref. [39]. Copyright 2021 Wiley-VCH Verlag GmbH & Co. KGaA, Weinheim, Germany.

**Figure 9 nanomaterials-12-01123-f009:**
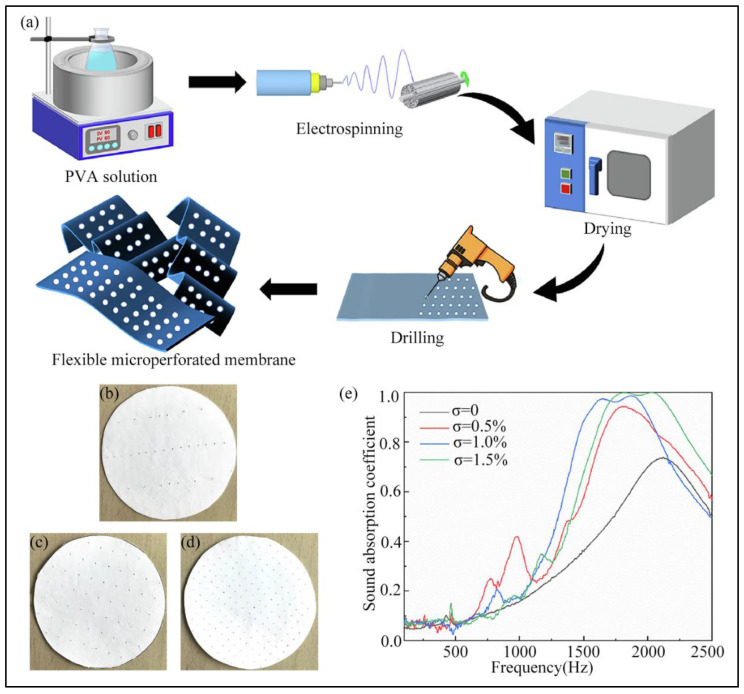
(**a**) The fabrication process of the flexible microperforated membrane; (**b**–**d**) Photographs of the microperforated samples with different perforation rates; (**e**) Sound absorption coefficient of the microperforated membranes with different perforation rates. Reprinted with permission from ref. [100]. Copyright 2022 Elsevier.

## Data Availability

Not Applicable.

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
