# Peer review of "Research Progress on Sound Absorption of Electrospun Fibrous Composite Materials"

_nanomaterials, 2022, doi:10.3390/nano12071123_

Round 1

Reviewer 1 Report

My remarks about the article "Research progress on sound absorption of electrospun fibrous composite materials"  are as follows:

The proposed review article, in which the focus has been on the performance of electrospun micro/nanofibers as sound absorbers, cited a total of 94 references. About  one third (32 references)  of the references belongs to 2020 and 2021. Although the manuscript includes a great deal of work with an established methodology, I recommend revision on the basis of the following:

Page 3, Table 1: I suggest replacing heading of the 4th column “Thickness of nanofibers” by “Thickness of nanofibrous structures”.

Page 3, Table 1: I suggest changing materials description for the reference [41] as “Polyvinylpyrrolidone (PVP) nonwoven mats of stacked nanofibrous layers.”

Page 5, Line 163: “When the diameter of the fibers is above approximately 3 μm, the specific surface area, flow resistivity, and the sound ab-163 sorption coefficient of the composite increase with the decrease of the fiber diameter.”. Is it above or below?

Page 6, Line 243-245: “Avossa et al. [41] obtained polyvinylpyrrolidone (PVP) non-woven mats by electrospinning with reduced thickness and excellent sound absorp-tion properties in the low and medium frequency range.” I, once again, suggest “Polyvinylpyrrolidone (PVP) nonwoven mats of stacked nanofibrous layers” definition fort he material description.

Page 6, Line 245-247: “The sound absorption coefficient is more significant than above 0.9 at 450 Hz, and its sound absorption performance can be continuously tuned by changing the mass.” More significant than what?

Page 8, Line 312-315: “Fibrous structures can absorb, reflect and transmit the incident sound waves simultaneously [65]. According to their compositions, acoustic fiber materials can be classified into natural, synthetic, and metal fibers. Natural and synthetic fibers are widely used due to their low price and superior sound absorption performance at high frequency.” This information about fiber materials should be given under heading 2.1. Materials section.

Page 11, Line 427-436: The paragraph between these lines is better given at the beginning of Section 3.

I think it would be better if sections 3.2 and 3.3 are combined.

Author Response

Dear Editor and Reviewers,

Thank you for allowing us to submit a revised manuscript, “Research progress on sound absorption of electrospun fibrous composite materials”, (No. 1641734) for publication in Nanomaterials. We appreciate the time and effort you and the reviewers dedicated to providing feedback on our manuscript. We have given a careful interpretation to the comments one by one. Those changes are marked up by using “Track Changes” and highlighted within the cover letter in light blue.

Point-by-Point Response to Comments

Reviewer #1

Page 3, Table 1: I suggest replacing heading of the 4th column “Thickness of nanofibers” by “Thickness of nanofibrous structures”.

=> We sincerely appreciate the reviewer for these comments.

We have replaced “Thickness of nanofibers” with “Thickness of nanofibrous structures” in the heading of the 4th column in Table 1 (Page 3, Line 109) in the revision.

Page 3, Table 1: I suggest changing materials description for the reference [41] as “Polyvinylpyrrolidone (PVP) nonwoven mats of stacked nanofibrous layers.”

=> We genuinely thank the reviewer for this.

We have revised the materials description from “Polyvinylpyrrolidone (PVP) non-woven mats” to “Polyvinylpyrrolidone (PVP) nonwoven mats of stacked nanofibrous layers” for the reference [41] in Table 1.

Page 5, Line 163: “When the diameter of the fibers is above approximately 3 μm, the specific surface area, flow resistivity, and the sound absorption coefficient of the composite increase with the decrease of the fiber diameter.”. Is it above or below?

=> Thank you very much for asking this question.

We have reread the original paper (Published by Akasaka, S. et al.; Reference [44] in the revision) to ensure the value of the fiber diameter. However, there is no specific value given in Reference [44]. They found the sound absorption performance of the samples is related to a specific fiber diameter, and they described it as the critical value (approximately 3 μm). In the revision, we have replaced the “approximately 3 μm” with “critical value” and added a sentence to explain it (Page 5, Line 178-184). The revision is as follow:

“They found that the sound absorption performance of the silica fibers is related to a critical value (approximately 3 μm) of the fiber diameter. When the fiber diameter is above the critical value, the sound absorption coefficient of the composite increase with the decrease of the fiber diameter in wide frequency range.”

Page 6, Line 243-245: “Avossa et al. [41] obtained polyvinylpyrrolidone (PVP) non-woven mats by electrospinning with reduced thickness and excellent sound absorp-tion properties in the low and medium frequency range.” I, once again, suggest “Polyvinylpyrrolidone (PVP) nonwoven mats of stacked nanofibrous layers” definition for the material description.

=> Thank you again for your suggestion.

We have also changed the material definition as “Polyvinylpyrrolidone (PVP) nonwoven mats of stacked nanofibrous layers” here (Page 7, Line 262-263) as we have done in Table 1 for Reference [41].

Page 6, Line 245-247: “The sound absorption coefficient is more significant than above 0.9 at 450 Hz, and its sound absorption performance can be continuously tuned by changing the mass.” More significant than what?

=> Thank you for your question.

We apologize for the confusion you find here. The point we were trying to make was that the sound absorption coefficient is higher than 0.9 at 450 Hz. We made a grammatical error. We have corrected this problem in the revision (Page 7, Line 265).

We are so sorry again for our mistake and sincerely thank you for noticing the problem.

Page 8, Line 312-315: “Fibrous structures can absorb, reflect and transmit the incident sound waves simultaneously [65]. According to their compositions, acoustic fiber materials can be classified into natural, synthetic, and metal fibers. Natural and synthetic fibers are widely used due to their low price and superior sound absorption performance at high frequency.” This information about fiber materials should be given under heading 2.1. Materials section.

=> Thank you for your kind suggestion.

We have carefully considered your suggestion about the adjustment of the paragraph. The 2.1 Materials section is focused on raw sources and additives for electrospinning, and the first paragraph in 3.2. section is related to the type of commercial sound absorption fibers now. There are some different between the two sections, so we think that the paragraph might be better to place at here.

We wish to have your understanding.

Page 11, Line 427-436: The paragraph between these lines is better given at the beginning of Section 3.

=> Thank you very much for your suggestion.

We have moved the paragraph at the end of Section 3. (Page 11, Line 427-436 in the original manuscript) has been moved to the beginning of Section 3. And we have adjusted the order of the sentences for better presentation. The rewritten part is as follow and shown in the revision (Page 7, Line 278-294).

“The poor sound absorption performance of porous sound absorption materials in the low and medium frequencies limits their applications. To broaden the applications of porous sound absorption materials, researchers combine them with electrospun micro/nanofibers. This method improves the sound absorption performance by expanding the contact area of the material with acoustic waves and enhances the material properties such as water resistance, high-temperature resistance, and mechanical strength. Furthermore, the addition of lightweight nanofibers will not vastly increase the weight and size of the sound absorption materials [43]. For instance, composite of electrospun micro/nanofiber with traditional foam and fiber sound absorption materials can increase the contact area be-tween the materials and sound waves, thus improving the sound absorption performance at the low and medium frequencies. Moreover, electrospinning micro/nanofibers with special structures can be prepared by refinement of electrospinning devices and material modification, which shows promising application prospects with a good sound absorption performance at low and medium frequencies.”

I think it would be better if sections 3.2 and 3.3 are combined.

=> Thank you for your suggestion.

We have discussed your suggestion for combining sections 3.2 and 3.3. However, we still prefer to describe these contents as two sections. Section 3.2 is focused on the research of composite of electrospun micro/nanofibers and commercial sound absorption fibers. And section 3.3 is intended to indicate the electrospun micro/nanofibers can also be used as sound absorption materials individually. Considering the content of section 3, the studies in section 3.3 are different to the research on the sound absorption performance of composites. So, it is valuable to describe these studies separately.

We hope you can accept our explanation about this question.

Reviewer 2 Report

This review reports recent studies on the sound absorption ability of electrospun fibers and their composite materials comprehensively. The authors focused on several factors, such as materials, fiber diameter, surface density, microstructure and thickness that are closed related with their sound absorption ability. When considering critical noise issue in our society, this work can be publishable at this journal. However, the following minor points should be clarified in the revision.

  1. On page 3: I would like to ask authors to define sound absorption coefficient (SAC) for a general reader.
  2. I would like to ask authors to include several sound absorption models that are developed up to now.
  3. Commercially available electrospun fiber composites for sound absorption materials?

Author Response

Dear Editor and Reviewers,

Thank you for allowing us to submit a revised manuscript, “Research progress on sound absorption of electrospun fibrous composite materials”, (No. 1641734) for publication in Nanomaterials. We appreciate the time and effort you and the reviewers dedicated to providing feedback on our manuscript. We have given a careful interpretation to the comments one by one. Those changes are marked up by using “Track Changes” and highlighted within the cover letter in light blue.

Point-by-Point Response to Comments

Reviewer #2

  1. On page 3: I would like to ask authors to define sound absorption coefficient (SAC) for a general reader.

=> We appreciate the reviewer for the suggestion.

We have added a definition about the sound absorption coefficient (SAC) on page 3, line 106-108 in the revision. And the definition is as follow:

“The sound absorption capacity of the materials can be reflected by SAC which is the ratio between the energy absorbed by the material and the total energy of incident sound wave.”

  1. I would like to ask authors to include several sound absorption models that are developed up to now.

=> Thank you for your insight.

We have added a paragraph to describe the acoustic models and introduced several commonly used acoustic models briefly on page 4, line 111-126 in the revision. And we have cited a few new reference (Reference [45-50]) in the description. The description is as follow:

“Acoustic models can be applied to theoretically evaluate the sound absorption properties of the materials, which are useful for directing the design of sound absorption materials. Nowadays, several acoustic models have been published, which are also been used for investigating the SAC of sound absorption materials by some researchers [39,44]. Until now, there are two types of acoustic models employed commonly [4]. One is the empirical model with few parameters, which is relatively easy to be established. The most representative empirical model is the Delany-Bazley model only related to the airflow resistivity of the materials, but it can only be employed for predicting the acoustic behaviors of porous materials in the frequency range of 250-4000 Hz [45,46]. In order to get more accurate predictions, some studies made some corrections to the Delany-Bazley model and developed several new models. For example, Miki proposed a modified expression known as Delany-Bazley-Miki model based on the Delany-Bazley model [47,48]. The other one acoustic model is the phenomenological model. Compared to the empirical models, it involves non-acoustical physical parameters and can improve the prediction accuracy. In terms of phenomenological model, the Johnson-Champoux-Allard (JCA) model is the most used one for describing the sound propagation in sound absorption porous materials [49,50].”

And the added references are as follow:

  1. Delany, M.E.; Bazley, E.N. Acoustical Properties of Fibrous Absorbent Materials. Appl. Acoust. 1970, 3, 105–116. https://doi.org/10.1016/0003-682X(70)90031-9.
  2. Liang, M.T.; Wu, H.G.; Liu, J.K.; Shen, Y.Q.; Wu, G.H. Improved Sound Absorption Performance of Synthetic Fiber Materials for Industrial Noise Reduction: A Review. J. Porous Mat. 2022, 1-24. https://doi.org/10.1007/s10934-022-01219-z.
  3. Miki, Y. Acoustical Properties of Porous Materials-Modifications of Delany-Bazley Models. J. Acoust. Soc. Jpn. (E). 1990, 11, 19–24. https://doi.org/10.1250/ast.11.19.
  1. Miki, Y. Acoustical Properties of Porous Materials-Generalizations of Empirical Models. J. Acoust. Soc. Jpn. (E). 1990, 11, 25–28. https://doi.org/10.1250/ast.11.25.
  2. Allard, J.F.; Champoux, Y. New Empirical Equations for Sound Propagation in Rigid Frame Fibrous Materials. J. Acoust. Soc. Am. 1992, 91, 3346-3353. https://doi.org/10.1121/1.402824.
  1. Chevillotte, F.; Perrot, C. Effect of the Three-Dimensional Microstructure on the Sound Absorption of Foams: A Parametric Study. J. Acoust. Soc. Am. 2017, 142, 1130–1140. https://doi.org/10.1121/1.4999058.

3. Commercially available electrospun fiber composites for sound absorption materials?

=> Thank you for your question.

We have tried for search the commercial sound absorption materials that containing electrospun micro/nanofibers. Unfortunately, we have not found any commercial electrospun fiber composites or reports related to them. Therefore, we consider that there are no commercially available electrospun fiber composites for sound absorption at present.

Other revision

We have revised a few details in the revision. Firstly, we cited six new references (Reference [45-50]) to introduce the acoustic models (Page 4, Line 111-126). It has been illustrated in the response at above. Secondly, we adjusted the order of the references according to the revision. Finally, we have completed the copyright information of the figures (Figures 3, 6(a-c), 6(d-f), 8).